# Rare Pancreatic/Peripancreatic Cystic Lesions Can Be Accurately Characterized by EUS with Through-the-Needle Biopsy—A Unique Pictorial Essay with Clinical and Histopathological Correlations

**DOI:** 10.3390/diagnostics13243663

**Published:** 2023-12-14

**Authors:** Maria Cristina Conti Bellocchi, Erminia Manfrin, Alessandro Brillo, Laura Bernardoni, Andrea Lisotti, Pietro Fusaroli, Alice Parisi, Sokol Sina, Antonio Facciorusso, Armando Gabbrielli, Stefano Francesco Crinò

**Affiliations:** 1Diagnostic and Interventional Endoscopy of Pancreas, Pancreas Institute, University of Verona, 37134 Verona, Italy; alessandro.brillo@studenti.univr.it (A.B.); laura.bernardoni@aovr.veneto.it (L.B.); armando.gabbrielli@univr.it (A.G.); stefanofrancesco.crino@aovr.veneto.it (S.F.C.); 2Department of Diagnostics and Public Health, Section of Pathology, University of Verona, 37134 Verona, Italy; erminia.manfrin@univr.it; 3Gastrointestinal Unit, Department of Medical and Surgical Sciences, Hospital of Imola, University of Bologna, 40026 Imola, Italy; lisotti.andrea@gmail.com (A.L.);; 4Department of Pathology and Diagnostics, University Hospital of Verona, 37126 Verona, Italy; alice.parisi@aovr.veneto.it (A.P.); sokol.sina@aovr.veneto.it (S.S.); 5Gastroenterology Unit, Department of Medical Sciences, University of Foggia, 00161 Foggia, Italy; antonio.facciorusso@unifg.it

**Keywords:** pancreatic cyst, endoscopic ultrasound, through-the-needle biopsy, pancreatic surgery, pancreatic cancer, fine-needle aspiration

## Abstract

Due to their aspecific macroscopic appearance, uncommon pancreatic cystic lesions (PCLs) are often misdiagnosed as mucinous lesions and improperly resected. We aimed to evaluate the endoscopic ultrasound (EUS)-guided through-the-needle biopsy (TTNB) capacity of the preoperative diagnosis of uncommon PCLs. Overall, 136 patients with PCLs who underwent EUS-TTNB between 2016 and 2022 were retrospectively identified. Common histotypes (e.g., IPMN, serous cystadenoma, and mucinous cystadenoma) were excluded and 26 (19.1%) patients (15 female, mean age 52.9 ± 10.4) were analyzed. The EUS findings, adverse events (AEs), and TTNB outcomes in uncommon PCLs were evaluated. The cysts histotype was accurately diagnosed by TTNB in 24/26 (92.3%) cases (seven cystic neuroendocrine tumors, four squamoid cysts, three acinar cells cystadenomas, two lymphoepithelial cysts, two mucinous non-neoplastic cysts, two bronchogenic cysts, two cystic lymphangiomas, one solid-pseudopapillary neoplasm, and one schwannoma). In the remaining two cases, lymphangioma was eventually diagnosed after resection. Surgery was performed in 15/26 (57.7%) patients. The mean follow-up of non-surgical patients was 32.5 months. One severe acute case of pancreatitis (3.8%) that required surgery occurred after EUS-TTNB. Uncommon pancreatic/peripancreatic lesions represent the 19.1% of PCLs in our series, with mainly benign histotypes. TTNB demonstrated a high diagnostic performance with a low rate of AEs in this setting, representing a reliable tool with which to avoid useless surgery.

## 1. Introduction

Pancreatic cystic lesions (PCLs) are commonly recognized due to the increasing use of cross-sectional imaging. PCLs have been categorized as neoplastic and non-neoplastic, including intraductal papillary mucinous neoplasms (IPMNs), mucinous cystic neoplasms (MCNs), serous cystic adenoma (SCA), and pseudocysts, with various malignant potential and different forms of management [1,2]. Moreover, several other uncommon cysts, often not requiring invasive management, can develop into or close to the pancreatic gland. Due to their rarity and macroscopic appearance as unilocular/oligocystic lesions that overlap with other PCLs, preoperative differential diagnosis may be challenging or even impossible. Indeed, due to the absence of specific features, these PCLs are often misdiagnosed as mucinous lesions and consequently unproperly resected, with the final diagnosis achieved upon the examination of the surgical specimen [3,4].

Endoscopic ultrasound (EUS) is considered the most accurate diagnostic procedure for the evaluation of PCLs, providing both a detailed description of their morphology and a guide for fine-needle aspiration (EUS-FNA). However, cytology on aspirated fluid is often inconclusive due to the scarce cellularity, and intra-cystic fluid marker analyses [e.g., carcinoembryonic antigen (CEA), amylase, or glucose] are not reliable for accurately distinguishing cyst histotypes [5].

In the last decade, a new microforceps device able to pass through a standard 19G EUS-FNA needle has been developed (Moray™ microforceps, US Endoscopy, Mentor, OH, USA). Once introduced into the cyst, the Moray™ microforceps enables a biopsy of the cystic wall to be performed, similar to standard biopsy forceps, with specimens containing both epithelium and stroma [6]. The procedure, called through-the-needle biopsy (TTNB), is technically feasible and relatively safe [7]. Both retrospective and prospective studies [8,9,10,11,12,13,14] have reported a histological yield ranging from 75% to 99.2% [15], and a specific cyst histotype has been recognized in 74% of cases [14]. Moreover, the interobserver agreement between expert pathologists has been demonstrated to be substantial [16]. However, all the above-mentioned studies have mainly described the performance of the Moray™ microforceps in common PCLs, but a description of TTNB specimens and procedural outcomes in rare PCLs is lacking.

The aim of this study was to evaluate the ability of the Moray™ microforceps to define uncommon PCL histotypes and the safety of the procedure in this subgroup of patients. Moreover, we provide a pictorial essay of the TTNB histological features of these rare lesions.

## 2. Materials and Methods

Uncommon PCLs were defined according to the European guidelines and included all cyst types other than IPMN, MCN, SCA, and pseudocyst [17].

Consecutive patients who underwent TTNB at the Endoscopic Unit of the Pancreas Institute of Verona between May 2016 and June 2022 were evaluated. Patients with common PCLs and those without a defined diagnosis at the time of the study were excluded. The local Ethic Committee approved the study.

Antibiotic prophylaxis was performed at the discretion of the endoscopists [18]. The TTNB procedures were performed by expert endosonographers as described below. A careful evaluation of the cystic lesions to identify the thickening of the wall or suspected nodules and contrast-harmonic EUS (CH-EUS) via the intravenous injection of 4.8 mL of SonoVue^®^ (Bracco, Milan, Italy) were performed to distinguish between vital tissue and debris or mucus plugs and guide the biopsy. The most suspicious area, a thickened wall, or a nodule according to the CH-EUS findings was targeted whenever possible. In the absence of suspicious areas, biopsies were performed randomly in the cyst wall. After puncturing the cyst with a 19G EUS-FNA needle, the stylet was removed, and a small amount of fluid was aspirated for biochemical analysis. The microforceps were then introduced through the needle, opened, gently pushed onto the cystic wall, closed, and pulled back until the ‘tent sign’ was seen. Two to four passes were performed with the aim of obtaining at least two macroscopically visible samples. The retrieved specimens were extracted from the microforceps’ jaws using the metal hook provided with the forceps, transferred between two 16 mm diameter paper disks (code 100623, Milestone srl, Sorisole, BG, Italy), and finally introduced into a cassette. Each specimen was placed separately in formalin vials. Samples were fixed in 10% formaldehyde solution and embedded in paraffin, sectioned at 5 μm, and stained with hematoxylin and eosin (H&E). Supplementary slides were prepared for histochemical and immunohistochemical staining. At the end of the TTNB procedure, the cystic fluid was, whenever possible, completely aspirated, centrifuged, and the sediment was smeared onto slides and processed according to Papanicolaou staining.

The adequacy rate was defined as the percentage of samples suitable for histological examination, and the accuracy was calculated against the final diagnosis, assessed on surgical specimens whenever available, or on a combination of clinical, imaging and cyto/histological features. Adverse events (AEs) were defined and classified according to the international lexicon [19].

Data collection was performed using patients’ charts revision. Moreover, all the patients included received telephone contact at the time of the study. To reduce the possibility of missing data, all the images of the cross-sectional radiology and EUS examinations, as well as the pathological specimens, were reviewed.

Patients’ characteristics were summarized by descriptive statistics [mean ± standard deviation (SD) or median with interquartile range (IQR) for continuous variables and frequency distributions for categorical variables]. Patients were grouped according to the lesion histotype, and the imaging/pathological findings, as well as the clinical outcomes, were described.

## 3. Results

During the study period, 136 patients underwent EUS-guided TTNB for the evaluation of PCLs. The indications for cyst puncture were those reported in international guidelines [2,17]. Ninety-one common PCLs (IPMNs, MCNs, SCAs, or pseudocysts) were excluded. Moreover, the diagnosis of 19 PCLs remained inconclusive after inadequate sampling. The patient selection process is summarized in the study flowchart (Figure 1).

Overall, 26/136 (19.1%) uncommon PCLs (15 female, mean age 52.9 ± 10.4) were studied. The TTNB samples were adequate in 25 out of 26 patients (adequacy rate of 96.2%), and accurate for diagnosis in 24 (accuracy rate of 92.3%). The mean size of lesions was 41.2 ± 21.2 mm. Uncommon PCLs were mainly unilocular (*n* = 17, 65.4%), had a thin wall (*n* = 15; 57.7%), and were variously situated in (*n* = 24) or close to the pancreatic gland (*n* = 2). The baseline characteristics and EUS findings are reported in Table 1.

Surgery was performed in 15/26 (53.8%) patients. The mean follow-up time in non-resected patients was 32.5 months. One (3.8%) severe acute case of pancreatitis requiring surgical necrosectomy occurred after the procedure in a patient with a squamoid cyst of the pancreatic duct (SCOP). The patients completely recovered without sequelae after intensive care unit hospitalization. Intraprocedural bleeding was documented by EUS in four patients and resolved uneventfully, without requiring intervention; these were classified as “incidents”, according to the ASGE lexicon [19]. The data of the included PCLs are resumed in Table 2 and described below. Table 3 summarizes the main EUS and pathological characteristics of the included cyst histotypes.

### 3.1. Cystic Neuroendocrine Tumor (cNET)

The published literature has reported that cNETs account for 13% of all pancreatic NETs [20]. Despite small cystic spaces in the context of a solid tumor being common in pancreatic NET, often assuming a “bull-eye” appearance with a central cyst and thickened wall, the complete cystic appearance is rare, and tendentially more indolent. cNETs may be sporadic or associated with multiple neuroendocrine neoplasia (MEN), or the von Hippel–Lindau or Wermer syndromes [21]. In our series, seven asymptomatic patients (four males, mean age 53 ± 6.8 years) presented a unilocular cyst (mean size 25.8 ± 6.7 mm) with a thickened wall (Figure 2A) located in the head, in the body, and in the tail of the pancreas in one, two and four cases, respectively. At CH-EUS, the hyperenhancement of thickened walls was observed (Figure 2B). The TTNB gathered histological specimens suitable for immunohistochemical stains in all cases. Numerous small round cells (Figure 2C) positive to chromogranin A and synaptophysin stains (Figure 2D) confirmed the neuroendocrine nature of the cysts. The evaluation of the proliferative index (Ki-67) of the TTNB specimens was possible in 6/7 (85.7%). All tumors were G1. Four out of seven patients underwent surgical resection that confirmed the preoperative diagnosis and the tumor grading. The remaining patients continued to be observed, with no changes over the follow-up period.

### 3.2. Squamoid Cyst of Pancreatic Ducts (SCOP)

SCOP is a very rare benign pancreatic cyst that has been recently described; it is usually well defined and unilocular, probably due to the dilation of a branch-duct with metaplastic squamous transformation [22,23]. In our series, four SCOPs were found, three in the head (22, 22, and 32 mm) and one in the body (40 mm) of the pancreas. They were regular, had a thin and fibrotic wall, and had a homogeneous anechoic content, except in one case, where a round-shaped avascular intracystic nodule was documented in both magnetic resonance imaging (MRI) and EUS (Figure 3A,B). In two cases, the levels of CEAs were elevated in the cyst fluid cytology. The TTNB retrieved fragments of a simple stratified squamous epithelium, without keratinization, lining the thin fibrous cyst wall (Figure 3C,D).

Due to the rarity of this cyst and the lack of information about its preoperative diagnosis, as well as the presence of thickened walls and increased CEA levels in the intracystic fluid analysis, two of the SCOPs were resected, given the persistent suspicion of the mucinous nature of the cyst despite the TTNB result. The surgery confirmed the diagnosis of the above-mentioned benign condition.

### 3.3. Acinar Cystic Transformation (ACT)

Acinar cystic transformation, also known as “acinar cell cystadenoma”, is an extremely rare pancreatic cystic neoplasm characterized by the acinar differentiation of epithelial cells without atypia and a good prognosis, although its potential for malignancy remains unclear [24]. Less than one hundred cases are described in the literature, being unilocular or multilocular and without a clearly preferred site within the pancreas. None of ACT’s specific clinical or radiological features are suspicious and a preoperative diagnosis is rarely formulated [25]. Three cases of ACT were found in our series; all were preoperatively diagnosed and each one was located in the head, body, or tail of the pancreas, with the appearance of an unilocular and oligocystic lesion. In two cases, a 35–37 mm large cyst was incidentally discovered, while the patient with the largest cyst (70 mm) complained of not-specific abdominal pain. In the histologic TTNB samples, acinar cells with a cuboidal shape containing round and basally oriented nuclei and eosinophilic periodic acid stain (PAS)-positive granules in its apical cytoplasm were observed. In the immunohistochemistry, positive staining for the markers of acinar cell differentiation, BCL10 and trypsin, was found (Figure 4). One patient underwent surgery, and an acinar epithelium with abortive acinar formations was observed in the surgical specimen. No dysplasia was found. The remaining two patients were scheduled for yearly follow-up with MRI. At the latest follow-up (19 and 12 months, respectively), no morphological changes were observed.

### 3.4. Cystic Lymphangioma (CL)

Pancreatic CLs are extremely uncommon (0.2% of PCLs), affecting typically young women; their growth is positively influenced by hormones (contraceptive, pregnancy, hyperprogesteronemia). Probably due to the progressive dilation of lymphatic vessels caused by insufficient drainage [26], CLs usually present as large and well-defined multicystic lesions with a thin septa, and less frequently have a unilocular or oligocystic appearance (Figure 5A). The intracystic viscous fluid seems chilous in the FNA, due to the triglycerides content [27,28] (Figure 5B). In our series, four middle-aged patients (three males) presented with a large cyst (mean size 60 mm) at different locations within the pancreas. In two out of four (50.0%) patients, no definitive characteristics of CLs were preoperatively found, either in the fluid cytology or TTNB specimens, and the diagnosis was reached after surgery. According to the microscopy, the two TTNB diagnostic specimens were composed of smooth muscle bundles and collagenous fibers interposed between slit-like vascular spaces lined by a flat endothelium; the D2-40 and CD31 immunolabelling was positive. Focally, small aggregates of lymphocytes were identified in the collagenous tissue (Figure 5C–F).

### 3.5. Lymphoepitelial Cyst (LeC)

Pancreatic LECs are very rare, comprising approximately 0.5% of all PCLs. They are usually observed in middle-aged men (5–6th decade) and have a slight preference for the body or tail of the pancreatic gland. The keratinized debris of LECs commonly results in a typically caseous appearance after FNA that can be misdiagnosed as MCNs, in addition to sometimes elevated CEA levels [29,30].

In this series, two 60-year-old males were found to be affected by large (48 and 60 mm) LECs; these had a unilocular and oligocystic appearance, were thin-walled and located in the pancreatic body, and had an inhomogeneous content, in one case assuming a “floating balls” appearance (Figure 6A) due to the sebaceous glands inside. The aspirated intracystic fluid was yellow-brown and thick, with rare squamous cells in a background of amorphous material. According to the microscopy, the TTNB specimen was largely composed of aggregates of mature lymphocytes lined by a squamous keratinizing epithelium, with sebaceous glands associated (Figure 6B–E). No surgery was performed, and both patients were followed for a mean time of 36 months without clinical/morphological changes.

### 3.6. Simple Mucinous Cysts (SMC)

SMCs, previously called “mucinous non-neoplastic cysts”, represent an emerging subset of PCLs, whose preoperative differential diagnosis from the well-known mucinous neoplasms (MCN and IPMN) remains very challenging. Despite the shared mucin-producing cells and elevated cyst fluid CEA levels, SMCs do not have malignant potential and can be differentiated from mucinous neoplasms by the lack of ovarian-type stroma, the absence of atypia, and no communication with the ductal system [31]. In our series, two SMCs, 18 mm in the tail and 40 mm in the head, were diagnosed with TTNB (Figure 7), the last being confirmed in the surgical specimen after duodenopancreatectomy.

### 3.7. Bronchogenic Cyst (BC)

Bronchogenic cysts are rare benign congenital anomalies originating from the endodermal foregut and are usually associated with maldevelopment. Normally, BCs are intra-thoracic, pulmonary or mediastinal, and having an abdominal location, both intra and extra-pancreatic, is extremely rare [32]. Surgical excision is usually recommended to establish the diagnosis, treat any symptoms, and prevent infective complications. Moreover, a remote risk of malignant transformation has been reported [33].

In our series, two middle-aged males presented a BC; one had a 60 mm intrapancreatic cyst located in the tail, and one had a peripancreatic 70 mm cyst located between the pancreas, left adrenal gland, posterior gastric wall, and diaphragm pillar.

Both the cysts were unilocular, thin-walled, and filled with a “starry sky” appearance due to multiple floating hyperechoic spots with comet-tail artifact (Figure 8A). According to the histology, the specimens were composed of a fibrous smooth muscle wall covered with a monolayer of “respiratory-type” epithelium with ciliated columnar cells and interspersed mucinous cells (Figure 8B–D). On the surgical specimens, the diagnosis was confirmed by the presence of seromucinous bronchogenic glands beneath the ciliated epithelium lining the cyst.

### 3.8. Solid Pseudopapillary Neoplasm (SPN)

Solid pseudopapillary neoplasms account for about 5% of all resected cystic tumors [34]. SPNs are more common in young females (female/male ratio 9:1, mean age 28 years) [35]. Cystic spaces are the consequence of degenerative changes, hemorrhages, and necrosis. Consequently, most SPNs show solid areas that are usually defined as standard solid tumors under EUS guidance. Complete cystic appearance is extremely rare. We observed one case of a unilocular 20 mm large cyst in the tail of the pancreas that was incidentally discovered in a 21-year-old female with a history of Hodgkin lymphoma 2 years earlier. TTNB revealed scattered aggregates of small cells, without significant atypia, organized in solid trabecular formations with a small hypocellular and myxoid stroma interposed. Tumor cells showed immunoreactivity for β-catenin, progesterone receptor and vimentin, and were negative to synaptophysin immunostaining (Figure 9). The patient underwent successful robotic enucleation with confirmation of the diagnosis. No recurrence occurred during the follow up.

### 3.9. Schwannoma (SWN)

SWNs originate from peripheral nerve sheath Schwann cells [36]. The pancreatic/peripancreatic localization of SWNs is extremely rare. Usually, SWNs arise from neighboring sites and secondarily involve the pancreatic gland, but more rarely can originate from pancreatic innervation [37]. SWNs are usually discovered incidentally as a painless mass in middle-aged patients. Malignant SWNs occur in 59% of patients with sporadic tumors [36]. SWNs are round or oval capsulated tumors with well-defined margins. Histologically, SWNs comprise two different areas: Antoni A and Antoni B. The first is composed of packed spindle-shaped cells that show immunohistochemistry positivity for protein S-100. The second area is occupied by hypocellular loose stroma and may be the subject of degenerative changes, such as cyst formation, hemorrhage, necrosis, and calcification. Cystic degeneration is related to larger sizes and mimics the whole spectrum of cystic pancreatic lesions. We observed one case of a large (103.5 mm) cystic unilocular SWN incidentally discovered on a transabdominal ultrasound and confirmed on a CT scan of a 40-year-old female (Figure 10A). According to the EUS, a unilocular cyst with a thickened irregular wall and corpuscolated fluid content was documented between the duodenal wall and the pancreatic head (Figure 10B). The TTNB specimens were composed of spindle cells with positive protein S-100 immunohistochemistry staining (Figure 10C–E). The lesion was subsequently surgically resected with confirmation of the diagnosis.

## 4. Discussion

EUS-TTNB is one of the most recent and promising techniques for the definition of PCLs, with excellent technical success (98.5%, CI 97.3–99.6%) and high accuracy (86.7%, CI 80.1–93.4) according to a recent metanalysis [15]. Several studies have compared TTNB to standard EUS-FNA, with conventional fluid analysis demonstrating the clear superiority of TTNB in risk stratification and its ability to provide a specific diagnosis [8,9,11,12,13,38,39,40,41,42,43,44,45,46,47,48]. Moreover, a study demonstrated that EUS-TTNB specimens are suitable for molecular analyses [49]. However, details of TTNB accuracy and its diagnostic yield in uncommon PLCs are lacking, specifically regarding the possibility of changing clinical management and avoiding unnecessary surgery.

To date, only surgical series or case reports of uncommon PLCs have been published in the literature, and the preoperative diagnosis is rarely formulated. The site and morphology are often overlapping, and no specific fluid analyses, including molecular diagnostics helping to differentiate different histotypes, are available. Consequently, pancreatic surgery, whose rate of complications and mortality is still considerable, is often improperly performed [3,4]. For this reason, the possibility of obtaining the diagnosis of a benign condition using EUS-TTNB becomes clinically relevant. To our knowledge, this is the first study describing the outcomes and histological features of TTNB specimens in uncommon PCLs.

In our series, TTNB showed an excellent adequacy (>96%) and accuracy (>92%) in diagnosing specific histotypes, with a high concordance with the surgical diagnosis in resected patients. The overall rate of AEs was low (3.8%), but physicians should be aware that severe AEs can occur after TTNB [7,50], as we observed in one patient who experienced severe acute pancreatitis. The occurrence of AEs after TTNB is a major concern among physicians. However, a recent large study was able to define three groups of patients with different levels of risk based on their cyst diagnosis, age, number of passes, and the complete aspiration of the cyst. In particular, IPMNs sampled with more than one pass were associated with AEs. In contrast, TTNB was safe in <64-year-old patients with lesions other than IPMNs sampled using two or fewer passes [7]. In the latter group, the risk of AEs was 1.4%, in agreement with the present study. Thus, this procedure should be reserved for selected cases and when an accurate diagnosis of the cyst histotype can influence patient management. Moreover, the cyst should be completely aspirated whenever possible, whereas the role of prophylactic measures such as rectal indomethacin/diclofenac administration or aggressive IV hydration is not yet established.

In the present study, a benign condition was accurately diagnosed by TTNB in 10 out of 12 cases, and surgery was subsequently avoided in 7/10 (70.0%). In the remaining three cases (30.0%), the management was influenced by the patient’s symptoms and/or wishes. Eventually, surgery confirmed the TTNB diagnosis in all cases. On the other hand, TTNB diagnosis allowed conditions requiring surgery such as SPN or SWN to be correctly managed; for these, surgical resection is internationally recommended [17] and they are associated with a positive long-term outcome. Moreover, a precise preoperative diagnosis for conditions with uncertain malignant potential, such as ACC and BC, allowed informed decisions to be shared between the patient and the physician.

In the present series, TTNB allowed cNET to be correctly diagnosed in all cases and provided a grading assessment in 85.7%; this is interestingly concordant with the surgical specimen in resected patients. Although cNETs tend to be biologically less aggressive than their solid counterparts, a 20% risk of malignancy has been reported, making the preoperative grading assessment equally relevant for risk stratification [2]. Meanwhile, in suspected solid pancreatic NET EUS tissue acquisition, and particularly in EUS-FNB, an optimal feasibility and a high rate of both diagnostic accuracy and correct grading classification compared to the surgical specimen have been documented [51]; in cNETs, the diagnostic yield of FNA ranges from 66.7% to 88.9% (targeting focal thickness or nodules on the walls), but to date, no reliable preoperative assessment of Ki67 has been demonstrated. It is likely that scraping or targeting the thickened wall using the needle might add an advantage, but the cellularity of the fluid retrieved is relatively low. The ability of TTNB to obtain samples of the epithelium lining the cyst and the tissue beyond might overcome this limitation [13,15]. Due to the rarity of the cystic appearance of pancreatic NET, however, to date, few data are available regarding the correct grading of cNET by TTNB, and the risk of misgrading due to the inhomogeneous expression of Ki67 in the cystic wall should be considered. In the future, it would be interesting to explore the possibility of performing the immunohistochemistry of new markers (i.e., DAXX/ATRX) that have demonstrated a strong correlation with histological features of aggressiveness [52]. Indeed, risk stratification is crucial also for the availability of less invasive locoregional treatments that could replace surgical resection in a limited subgroup of patients [53,54].

In the present series, the most challenging diagnosis seemed to be CLs, with only two out of four cases (50.0%) diagnosed with TTNB. Since the endothelial cells of the outer membrane of pancreatic CLs do not normally undergo epithelial differentiation, several staining agents are unsuitable, making the preoperative diagnosis of this condition extremely challenging [55]. Nevertheless, in addition to clinical information and radiological details, EUS provided a better morphological description and allowed both materials for fluid analysis and histological evaluation to be obtained, increasing the possibility of reaching a diagnosis. Due to the challenging diagnosis of CLs using the TTNB specimens or fluid cytology, other through-the-needle techniques, such as confocal endomicroscopy, could be considered.

Several limitations of this study should be addressed. First, the relatively small sample size of 136 patients, with only 26 analyzed for uncommon pancreatic cystic lesions, may impact the generalizability of the findings. A larger sample size would strengthen the study’s applicability. Second, the inclusion of consecutive patients undergoing TTNB at a specific institution may introduce selection bias. Third, we acknowledge that the histotype of 19 lesions remained undefined and we cannot be sure that other uncommon PCLs were missed, thus reducing the overall diagnostic adequacy and accuracy of TTNB in this setting. Fourth, the exclusion of patients with undefined lesions and common histotypes, such as IPMN, SCA, and MCN, may limit the study’s applicability to a broader population where these lesions are prevalent. Finally, the surgical confirmation of the cyst histotypes was available in only 53.8% of patients, thus carrying a potential bias of accuracy overestimation.

## 5. Conclusions

Preoperative diagnosis is essential to prevent unnecessary surgical intervention to treat benign uncommon cysts mimicking mucinous lesions. This is the first series reporting the outcomes of TTNB in this setting and providing a pictorial essay on EUS and the histological features that may serve as a guide for both endosonographers and pathologists in recognizing these rare lesions.

## Figures and Tables

**Figure 1 diagnostics-13-03663-f001:**
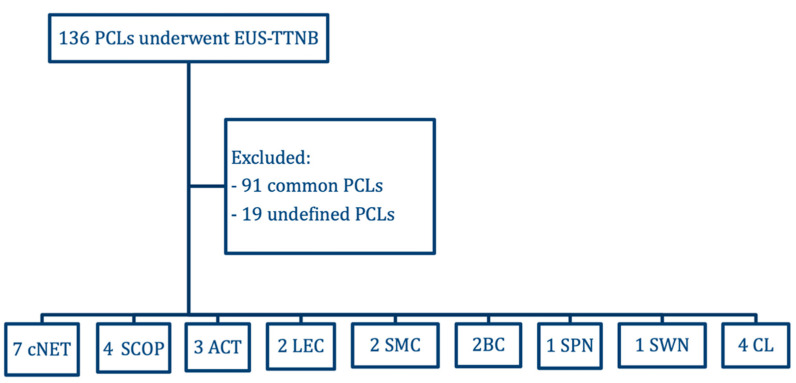
Flowchart of the study. EUS-TTNB, endoscopic ultrasound-guided through-the-needle biopsy; PCLs, pancreatic cystic lesions; cNET, cystic neuroendocrine tumors; SCOP, squamoid cyst of pancreatic duct; ACT, acinar cystic transformation; LEC, lymphoepithelial cyst; SMC, simple mucinous cyst; BC, bronchogenic cyst; SPN, solid pseudopapillary neoplasm; SWN, schwannoma; CL, cystic lymphangioma.

**Figure 2 diagnostics-13-03663-f002:**
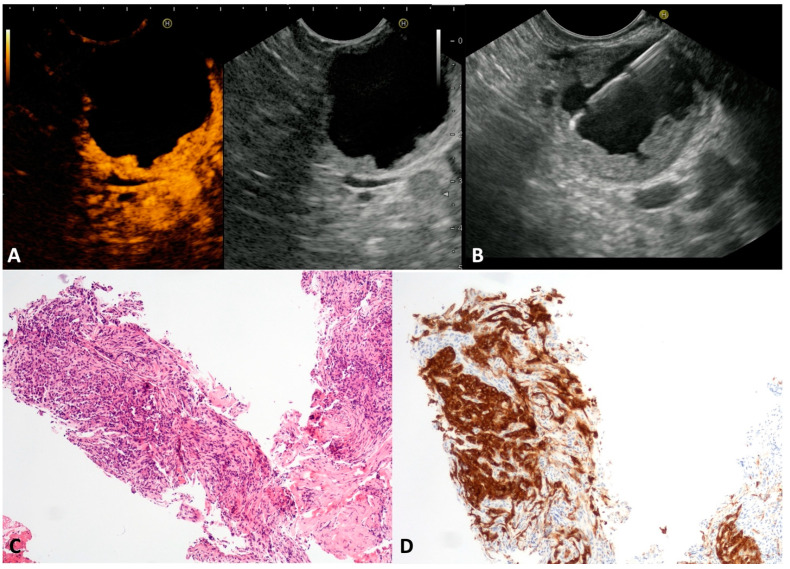
Cystic neuroendocrine neoplasm. Contrast-harmonic endoscopic ultrasound showing hyper-enhanced walls of an unilocular cyst (**A**). Endoscopic ultrasound-guided through-the-needle biopsy targeting the thickened walls (**B**). The cyst wall is almost completely composed of small, tightly packed epithelial cells (**C**) that stain intensely for the neuroendocrine marker Synaptophysin (**D**). Hematoxylin–eosin original magnification ×100 (**C**). Synaptophysin original magnification ×100 (**D**).

**Figure 3 diagnostics-13-03663-f003:**
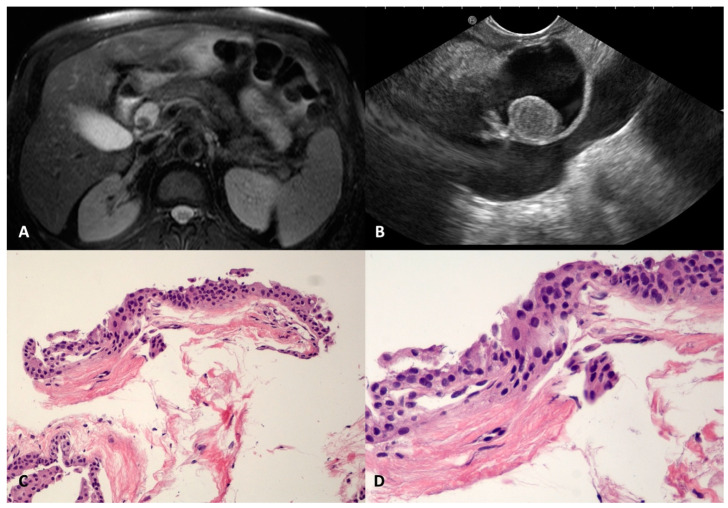
Squamoid cyst of pancreatic ducts. T2-weighted magnetic resonance imaging of a pancreatic cyst located in the head, with a nodule inside (**A**). The same cyst on an endoscopic ultrasound scan containing round vegetation that resulted in avascular at contrast-harmonic evaluation (**B**). The fibrous thin wall of the cyst is lined by stratified epithelium (**C**) without atypia and keratinization (**D**). Hematoxylin–eosin original magnification ×100 (**C**), ×200 (**D**).

**Figure 4 diagnostics-13-03663-f004:**
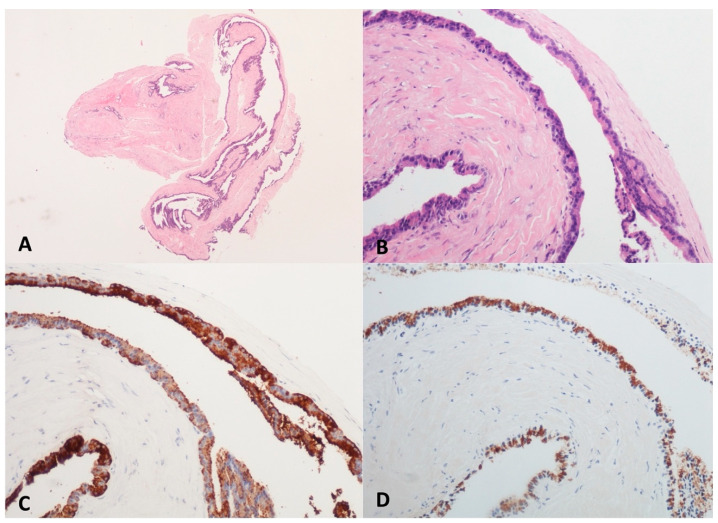
Acinar Cystic Transformation. The whole mounted section of the multilocular cyst with fibrous wall (**A**). The lining epithelium is composed of cuboidal cells with abundant eosinophilic and granular cytoplasm, organized both in monolayer and acinar aggregates (**B**). The acinar markers BCL10 (**C**) and trypsin (**D**) are detected in epithelial cells. Hematoxylin–eosin original magnification ×40 (**A**), ×100 (**B**); BCL10 original magnification ×100 (**C**); Trypsin original magnification ×100 (**D**).

**Figure 5 diagnostics-13-03663-f005:**
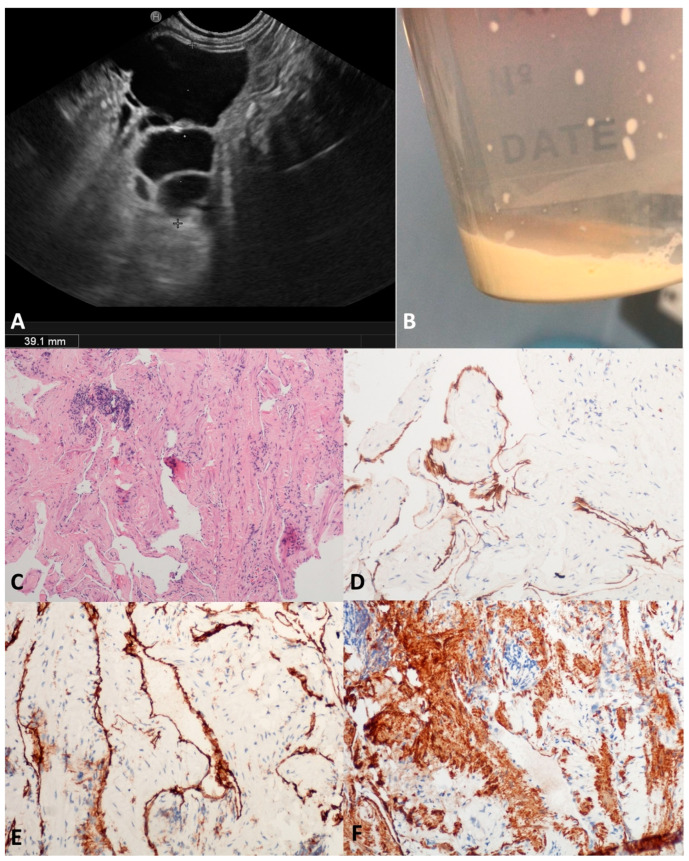
Cystic lymphangioma. Endoscopic ultrasound revealed an irregularly shaped multilocular cyst close to the pancreatic parenchyma, with a thin wall and septa (**A**). The aspirated fluid appeared as thick “milky” white-yellowish fluid (**B**). Numerous narrow lymphatic vessels lined by a cuboidal epithelium were separated by bundles of smooth muscle with few aggregates of lymphocytes (**C**). Immunolabelling for endothelial markers D2-40 (**D**) and CD31 (**E**). Bundles of smooth muscle actin (SMA) positive cells (**F**). Hematoxylin–eosin original magnification ×100 (**C**); D2-40 original magnification ×200 (**D**); CD31 original magnification ×200 (**E**); SMA original magnification ×200 (**F**).

**Figure 6 diagnostics-13-03663-f006:**
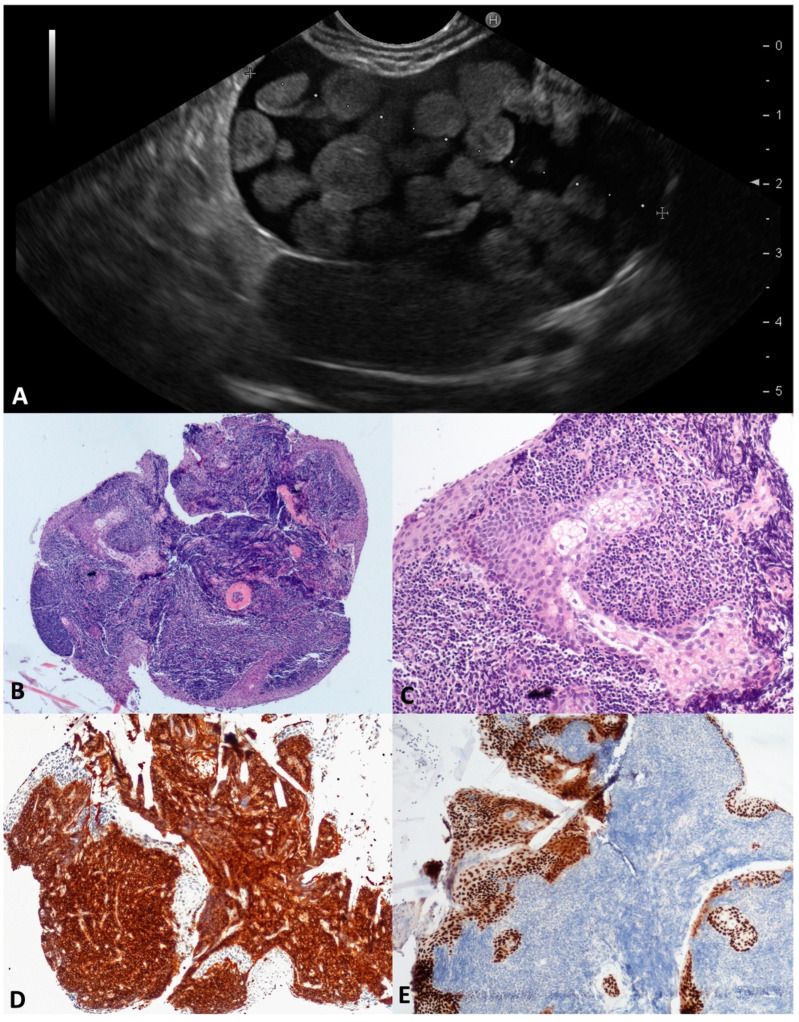
Lymphoepithelial cyst. During the endoscopic ultrasound, a large cystic lesion with a regular wall and central thickened septum containing multiple round, echoic, and avascular floating balls after contrast injection was documented (**A**). The whole mounted section of the cyst wall forceps biopsy (**B**). Large aggregates of lymphoid cells are separated by squamous epithelial cells with sebaceous glands (**C**). Lymphocytes are highlighted by CD45 (**D**) and squamous cells by P63 (**E**) immunostaining, respectively. Hematoxylin–eosin original magnification ×40 (**B**), ×200 (**C**); CD45 original magnification ×100 (**D**); P63 original magnification ×100 (**E**).

**Figure 7 diagnostics-13-03663-f007:**
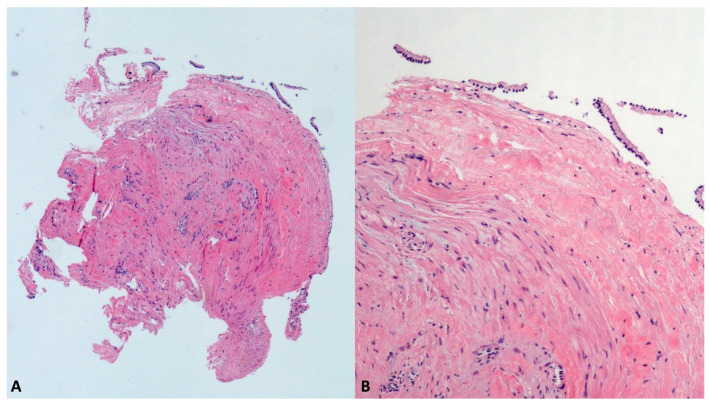
Simple mucinous cyst. The whole fully mounted section of the cyst wall forceps biopsy (**A**). At higher magnification, the thick fibrous cist wall is lined with a monolayer of mucinous epithelial cells, without atypia (**B**). Hematoxylin–eosin original magnification ×40 (**A**), ×200 (**B**).

**Figure 8 diagnostics-13-03663-f008:**
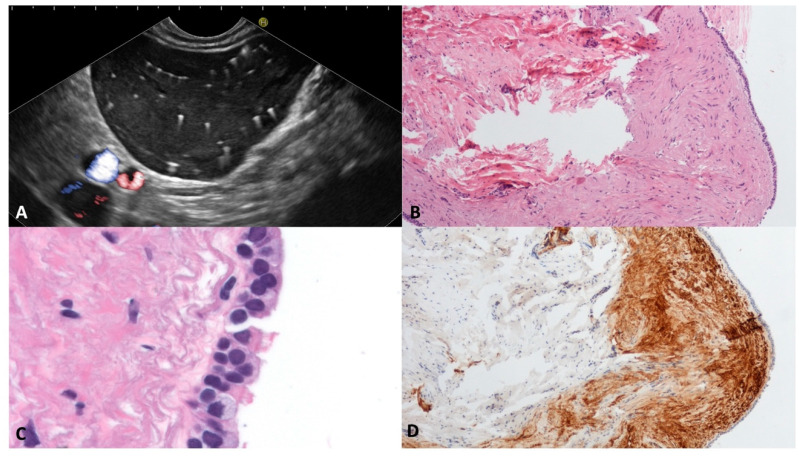
Bronchogenic cyst. Typical appearance of bronchogenic cyst on endoscopic ultrasound. The content of the cyst is similar to the liver parenchyma, with multiple white spots with a comet-tail artifact (**A**). The cyst wall biopsy shows a thick layer of eosinophilic, dense, hypocellular tissue beneath the pseudostratified epithelium (**B**). The epithelium lining the cyst wall is composed of ciliated and goblet cells (**C**). The subepithelial connective tissue is rich in smooth muscle fibers, which are Calponin-positive (**D**). Hematoxylin–eosin original magnification ×100 (**B**), ×400 (**C**); Calponin original magnification ×100 (**D**).

**Figure 9 diagnostics-13-03663-f009:**
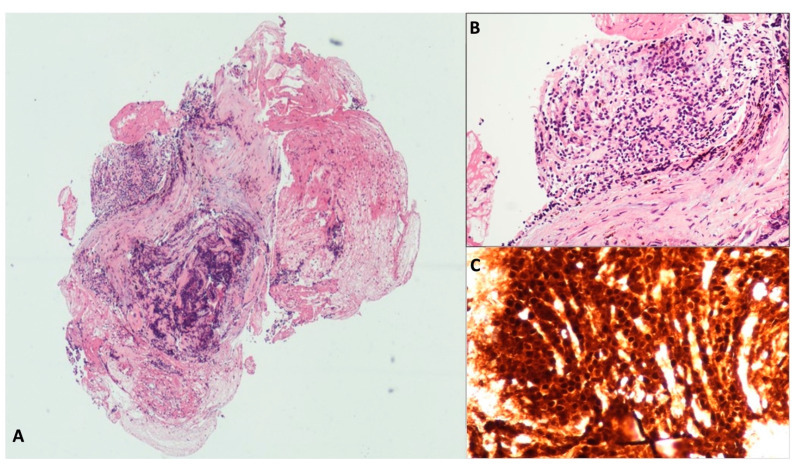
Solid pseudopapillary tumor. The whole mounted section of the cyst wall tissue fragment shows an altered histological architecture with an area of crash artefact caused by the forceps sampling procedure (**A**). In an area with better preserved morphology, the epithelial cells are round, small and tightly packed (**B**). Immunolabelling for β-catenin with aberrant nuclear positivity is conclusive for the diagnosis (**C**). Hematoxylin–eosin original magnification ×40 (**A**), ×200 (**B**); β-catenin original magnification ×200 (**C**).

**Figure 10 diagnostics-13-03663-f010:**
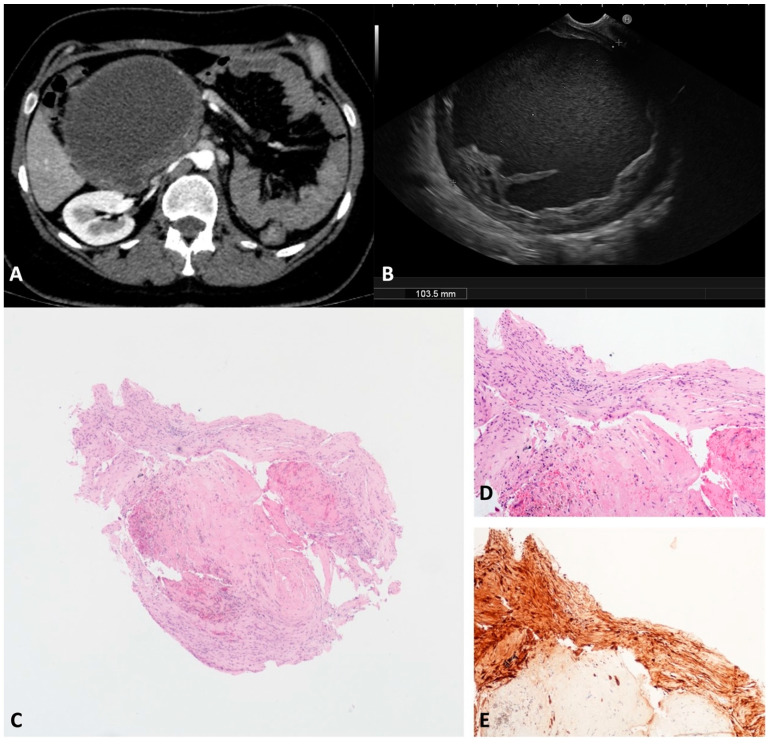
Schwannoma. Computed tomography appearance of a large cystic lesion located between the second/third portion of the duodenum and the pancreatic head that appeared compressed and dislodged (**A**). On the endoscopic ultrasound, the lesion appeared as a round unilocular cyst with smooth borders and an irregular thickened wall (**B**). The whole mounted section of the cyst wall biopsy shows no epithelium lining the cyst (**C**). A well visible fascicle of spindle cells depicts the peripheral profile of the biopsy (**D**). Diffuse and intense immunolabelling for S100 in spindle cells (**E**). Hematoxylin–eosin original magnification ×40 (**C**), ×100 (**D**). S100 original magnification ×100 (**E**).

**Table 1 diagnostics-13-03663-t001:** The baseline characteristics of the patients included and endoscopic ultrasound findings.

	*N* = 26
Mean age of patients	mean age 52.9 ± 10.4
Sex	
Male/female	11/15
Clinical onset	
Asymptomatic	18 (69.2%)
Abdominal pain	6 (23.1%)
Dispepsia	2 (7.7%)
Acute pancreatitis	0 (0%)
Site	
Head	7 (26.9%)
Body	8 (30.8%)
Tail	9 (34.6%)
Extrapancreatic	2 (7.7%)
Mean size of the cyst, mm ± SD	41.2 ± 21.2 mm
Type of lesion	
Unilocular	17 (65.4%)
Oligocystic	9 (34.6%)
Walls	
Thin	15 (57.7%)
Thickened	11 (42.3%)
Mean needle passes	1
Mean TTNB passes	2.9
Complete aspiration of the cyst	13 (50%)

SD, standard deviation; TTNB, through-the-needle biopsy.

**Table 2 diagnostics-13-03663-t002:** Detailed findings of uncommon pancreatic cystic lesions.

	cNET	SCOP	ACT	CL	LeC	SMC	BC	SPN	SWN
*N*° of cases	7	4	3	4 *	2	2	2	1	1
Mean size (mm), mean ± SD	25.8 ± 6.7	29.0 ± 7.5	47.6 ± 15.8	57.5 ± 8.3	56.5 ± 8.5	29.0 ± 11.0	65.0 ± 5.0	25.0	103.5
Location, *n* (%)									
Head	1 (14.3%)	2 (50%)	1 (33.3%)	2 (50%)	1 (50%)	1 (50%)	0 (0%)	1 (100%)	1 (100%)
Body	2 (28.6%)	2 (50%)	1 (33.3%)	1 (25%)	1 (50%)	0 (0%)	0 (0%)	0 (0%)	0 (0%)
Tail	4 (57.1%)	0 (0%)	1 (33.3%)	1 (25%)	0 (0%)	1 (50%)	2 (100%)	0 (0%)	0 (0%)
Lesions wih thickned wall, *n* (%)	7/7 (100%)	1/4 (25%)	1/3 (33%)	0/4 (0%)	0/2 (0%)	0/2 (0%)	0/2 (0%)	0/1 (0%)	1/1 (100%)
Adverse event	0	1 °	0	0	0	0	0	0	0
Malignant potential	yes	no	uncertain	no	no	no	uncertain	yes	yes
Surgery, *n* (%)	4/7 (57%)	2/4 (50%)	1/3 (33%)	2/4 (50%)	0/2 (0%)	1/2 (50%)	2/2 (100%)	1/1 (100%)	1/1 (100%)

* Two cases diagnosed after surgery. ° pancreatitis. cNET, cystic neuroendocrine tumors; SCOP, squamoid cyst of pancreatic duct; ACT, acinar cystic transformation; CL, cystic lymphangioma; LeC, lymphoepithelial cyst; SMC, simple mucinous cyst; BC, bronchogenic cyst; SPN, solid pseudopapillary neoplasm; SWN, schwannoma; SD, standard deviation.

**Table 3 diagnostics-13-03663-t003:** Endoscopic ultrasound, contrast-harmonic ultrasound, pathological and immunohistochemistry features of uncommon pancreatic/peripancreatic cystic lesions.

Cyst Histotype	EUS Features	CH-EUS Features	Main Pathological Findings at TTNB	IHC
cNET	Unilocular with thickened wall	Arterial-phase hyper-enhanced wall	Uniform size and shape tumor cells with round, nucleus and finely dispersed chromatin	Cell positivity for Synaptophysin, Chromogranin A, and Cytokeratin 8/18
SCOP	Unilocular, often with thin walls	Not specific	Simple stratified squamous epithelium, without keratinization, lining thin fibrous cyst wall	Not specific
ACT	Unilocular or oligocystic, can have thickened walls or vegetations	Avascular content	Acinar cuboidal cells. Round, basally oriented nuclei and eosinophilic, periodic acid stain-positive granules in apical cytoplasm	Cell positivity for BCL10 and trypsin
CL	Usually, multicystic with thin walls. Aspirated fluid can appear as milky due to triglyceride content	Not specific	Lymphatic vessels lined by cuboidal epithelium separated by bundles of smooth muscle with few aggregates of lymphocytes	Endothelium cells positive for D2-40 and CD31. Muscle cells positive for smooth muscle actin (SMA)
LeC	Unilocular or oligocystic with thin walls and inhomogeneous content, usually located in the pancreatic body	Avascular content	Aggregates of mature lymphocytes lined by squamous keratinizing epithelium, with sebaceous glands associated	Lymphocytes positive for CD45. Squamous cells positive for P63
SMC	Unilocular with thin walls	Not specific	Fibrous cist wall lined by a monolayer of mucinous epithelial cells, without atypia	Not specific
BC	Unilocular with thin walls and “starry sky” appearance of content. Usually located adjacent to the diaphragm pillar	Avascular content	Eosinophilic, hypocellular connective tissue rich in smooth muscle fibers beneath a pseudostratified epithelium composed by ciliated and goblet cells	Muscle fibers positive for Calponin
SPN	Unilocular with thickened and irregular wall. Inhomogeneous solid/cystic content	Vascularized wall with avascular content	Scattered aggregates of small cells organized in solid-trabecular formations with a small hypocellular and myxoid stroma interposed	Tumor cells positive for β-catenin, progesterone receptor and vimentin
SWN	Unilocular with thickened and irregular wall. Inhomogeneous content	Vascularized wall with avascular content	Fascicles of spindle cells	Spindle cells positive for protein S-100

EUS, endoscopic ultrasound; CH-EUS, contrast-harmonic endoscopic ultrasound; TTNB, through-the-needle biopsy; IHC, immunohistochemistry; cNET, cystic neuroendocrine tumors; SCOP, squamoid cyst of pancreatic duct; ACT, acinar cystic transformation; CL, cystic lymphangioma; LeC, lymphoepithelial cyst; SMC, simple mucinous cyst; BC, bronchogenic cyst; SPN, solid pseudopapillary neoplasm; SWN, schwannoma.

## Data Availability

Data is contained within the article.

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
