# Peer review of "Rare Pancreatic/Peripancreatic Cystic Lesions Can Be Accurately Characterized by EUS with Through-the-Needle Biopsy—A Unique Pictorial Essay with Clinical and Histopathological Correlations"

_diagnostics, 2023, doi:10.3390/diagnostics13243663_

Round 1

Reviewer 1 Report

Comments and Suggestions for Authors

I read with interest the paper by Maria Cristina Conti Bellocchi and colleagues. The study addresses the diagnostic challenges associated with uncommon pancreatic cystic lesions, offering valuable insights into the application of endoscopic ultrasound-guided through-the-needle biopsy (EUS-TTNB). This innovative approach contributes to the existing literature on pancreatic lesions. The paper is centered around a clear and specific research question, evaluating the diagnostic capability of EUS-TTNB in establishing preoperative diagnoses for uncommon pancreatic cystic lesions. I appreciate its detailed analysis of various histotypes within this category.

The study places significant emphasis on the importance of avoiding unnecessary surgeries, a critical aspect of its findings. If EUS-TTNB proves to be a reliable tool for accurate diagnosis, it could have substantial clinical implications, preventing unnecessary interventions and associated risks.

Several limitations should be acknowledged and addressed whenever possible.

·       The methodology briefly mentions the use of descriptive statistics for data analysis. A more comprehensive explanation of the statistical methods, including any subgroup analyses or adjustments for confounding variables, would enhance the analytical approach.

·       The retrospective nature of the study introduces potential biases, as data is gathered from past cases, limiting researchers' control over the data collection process. The relatively small sample size of 136 patients, with only 26 analyzed for uncommon pancreatic cystic lesions, may impact the generalizability of the findings. A larger sample size would strengthen the study's applicability. Furthermore, the inclusion of consecutive patients undergoing TTNB at a specific institution may introduce selection bias. The exclusion of patients with undefined lesions could also affect the generalizability of the findings, as these cases are common in real-world clinical settings. This aspect should be explicitly addressed in the discussion. The exclusion of common histotypes, such as IPMN, serous cystadenoma, and mucinous cystadenoma, may limit the study's applicability to a broader population where these lesions are prevalent Ninety-one common PCLs (IPMNs, MCNs, SCAs, or pseudocysts) were excluded and 19 PCLs remained inconclusive after inadequate sampling. Authors should explain how they dealt with missing data in the methodology.

·       While the description of the TTNB procedure is comprehensive, additional details, such as the number of biopsy passes, specific criteria for targeted biopsies, and the role of contrast-harmonic EUS in guiding the biopsy, could enhance the reproducibility of the study.

·       Transparent reporting of adverse events, including a case of severe acute pancreatitis, is a noteworthy aspect of the study. This information is essential for clinicians and researchers evaluating the safety profile of the EUS-TTNB procedure and should be implemented in the discussion

·       I really appreciated images from contrast-harmonic endoscopic ultrasound and hematoxylin-eosin original magnification. I would only add a summary table with histopathology types and main characteristics (pathology and EUS) to provide the reader with a more comprehensive ideas of EUS-TTNB potential and pancreatic pathology disease (cystic neuroendocrine tumors, squamoid cysts, acinar cells cystadenomas, lymphoepithelial cysts, mucinous non-neoplastic cysts, bronchogenic cysts, cystic lymphangiomas, solid-pseudopapillary neoplasms, and schwannoma).

Comments on the Quality of English Language

It can be improved.

Author Response

1)       The methodology briefly mentions the use of descriptive statistics for data analysis. A more comprehensive explanation of the statistical methods, including any subgroup analyses or adjustments for confounding variables, would enhance the analytical approach.

RE: Thank you for your suggestion. This is a descriptive case series with a relatively small number of patients and only descriptive statistics has been applied. To better explain this point, we specified that analyzed lesions were grouped according to histotype, and imaging/pathological features and clinical outcomes were described.

2)       The retrospective nature of the study introduces potential biases, as data is gathered from past cases, limiting researchers' control over the data collection process. The relatively small sample size of 136 patients, with only 26 analyzed for uncommon pancreatic cystic lesions, may impact the generalizability of the findings. A larger sample size would strengthen the study's applicability. Furthermore, the inclusion of consecutive patients undergoing TTNB at a specific institution may introduce selection bias. The exclusion of patients with undefined lesions could also affect the generalizability of the findings, as these cases are common in real-world clinical settings. This aspect should be explicitly addressed in the discussion. The exclusion of common histotypes, such as IPMN, serous cystadenoma, and mucinous cystadenoma, may limit the study's applicability to a broader population where these lesions are prevalent Ninety-one common PCLs (IPMNs, MCNs, SCAs, or pseudocysts) were excluded and 19 PCLs remained inconclusive after inadequate sampling. Authors should explain how they dealt with missing data in the methodology.

RE: Thank you for your comment. We addressed the study limitations you raised up in the discussion section and we stated how data collection was performed in the methods section.

3)       While the description of the TTNB procedure is comprehensive, additional details, such as the number of biopsy passes, specific criteria for targeted biopsies, and the role of contrast-harmonic EUS in guiding the biopsy, could enhance the reproducibility of the study.

RE: Thank you. We added further procedural details in the Methods section.

4)       Transparent reporting of adverse events, including a case of severe acute pancreatitis, is a noteworthy aspect of the study. This information is essential for clinicians and researchers evaluating the safety profile of the EUS-TTNB procedure and should be implemented in the discussion

RE: We agree with your comment and we implemented the discussion part concerning TTNB-related AEs

5)       I really appreciated images from contrast-harmonic endoscopic ultrasound and hematoxylin-eosin original magnification. I would only add a summary table with histopathology types and main characteristics (pathology and EUS) to provide the reader with a more comprehensive ideas of EUS-TTNB potential and pancreatic pathology disease (cystic neuroendocrine tumors, squamoid cysts, acinar cells cystadenomas, lymphoepithelial cysts, mucinous non-neoplastic cysts, bronchogenic cysts, cystic lymphangiomas, solid-pseudopapillary neoplasms, and schwannoma).

RE: Thank you for your suggestion. We added the “Table 3” with morphological and pathological findings in the Result section

Reviewer 2 Report

Comments and Suggestions for Authors

The manuscript entitled:" Rare pancreatic/peripancreatic cystic lesions can be accurately characterized by EUS with through-the-needle biopsy. A unique  pictorial essay with clinical and histopathological correlations" focuses on the identification of a novel diagnostic alghoritm able to identify rare peripancreatic cystic lesions is technically correct. Accordingly, few minor considerations should be approached to improve the readibility on this journal

- In the introduction section, please, could the authors overview the main issues in diagnostic sessions of peripancreatic lesions

- in the methodological section, please, could the authros provide a table with IHC data integrated morphological evaluation of this setting?

- In the discussion section, please, could the authors discuss the role of molecular techniques in this setting?

Comments on the Quality of English Language

minor english editing

Author Response

- In the introduction section, please, could the authors overview the main issues in diagnostic sessions of peripancreatic lesions

RE: thank you for your comment. As we stated in the introduction, morphological characteristics of uncommon cysts overlap with mucinous lesions. Due to the absence of specific features, the differentiation is often impossible based on preoperative examinations. We further underlined this point in the text.

- in the methodological section, please, could the authros provide a table with IHC data integrated morphological evaluation of this setting?

RE: Thank you for your suggestion. We added the “Table 3” with morphological and ICH findings in the Result section

- In the discussion section, please, could the authors discuss the role of molecular techniques in this setting?

RE: to our knowledge, there are not specific molecular diagnostics for rare pancreatic lesions. We added a comment on this point in the text

Reviewer 3 Report

Comments and Suggestions for Authors

This is a single site observational study in evaluating the TTNB technique in diagnosing uncommon PCLs which is very relevant in management of uncommon PCLs. I would like to take this opportunity to applaud the work done in this manuscript. Just a few comments.

Abstract- Clear and concise.

Introduction- Clear introduction.

Methods- Good explanation on the TTNB technique.

Results

1.      Could the authors specify in Table 2, wall thickness was based on which guideline and the reason of using percentage in categorising the wall thickness.

2.      Could the authors include what was the follow up guideline for the patients who did not have surgery. In line 230, 2 patients were followed up for 19 and 12 months without morphological changes, does that mean the patient had further TTNB? If so, how often?

3.      2 CL patients were only diagnosed after surgery. In your inclusion criteria, patients with undefined PCLs after inadequate sampling were excluded. Could the authors explain why these 2 patients who were undiagnosed with TTNB technique were included.

4.      Follow up question from point 3, if these 2 patients where diagnosis were achieved after surgery, could the authors clarify if the 19 excluded patients had diagnosis after treatment/ surgery as they could provide important information for this paper.

Author Response

  1. Could the authors specify in Table 2, wall thickness was based on which guideline and the reason of using percentage in categorising the wall thickness.

RE: Thank you for your comment. Wall thickness was based on European guidelines (Gut. 2018, 67, 789 – 804), despite there is not a clear definition of “thickened wall” in any of the available guidelines. In the table 2, we reported the rate of lesions with thickened wall, not the percentage of wall thickness in the single lesion. Sorry for the lack of clarity. We changed the label of the column in “Lesions with thickened wall, n (%)”

  1. Could the authors include what was the follow up guideline for the patients who did not have surgery. In line 230, 2 patients were followed up for 19 and 12 months without morphological changes, does that mean the patient had further TTNB? If so, how often?

RE: Due to the rarity of these lesions, there are no guidelines for the follow-up schedule. The two patients you referred to were diagnosed to have an acinar cystic transformation on TTNB sampled and were followed up using MRI yearly. We specified this point in the text. No further TTNB were repeated because the index procedure was already adequate.

  1. 2 CL patients were only diagnosed after surgery. In your inclusion criteria, patients with undefined PCLs after inadequate sampling were excluded. Could the authors explain why these 2 patients who were undiagnosed with TTNB technique were included.

RE: Sorry for the lack of clarity. We excluded those lesions that remained “undefined” regardless the results of TTNB (line 82). That two patients who you refer to, eventually received a diagnosis after surgery and, therefore, had a “defined” lesion despite TTNB was inconclusive. We modified the manuscript to clarify this point.

  1. Follow up question from point 3, if these 2 patients where diagnosis were achieved after surgery, could the authors clarify if the 19 excluded patients had diagnosis after treatment/ surgery as they could provide important information for this paper.

RE: No, those 19 patients had not a definitive diagnosis at the time of the study, and this is the reason why they were excluded

Round 2

Reviewer 1 Report

Comments and Suggestions for Authors

No further comments

Comments on the Quality of English Language

Minor edits